# Modelling a century of soil redistribution processes and carbon delivery from small watersheds using a multi-class sediment transport model

Florian Wilken[1,2,3], Peter Fiener[1], Kristof Van Oost[4]

[1]Institute for Geography, Universität Augsburg, Augsburg, D-86159, Germany
[2]Chair of Soil Protection and Recultivation, Brandenburg University of Technology Cottbus-Senftenberg, Cottbus, D-03046, Germany
[3]Institute of Soil Landscape Research, Leibniz-Centre for Agricultural Landscape Research (ZALF) e.V., Müncheberg, D-15374, Germany
[4]Earth & Life Institute/TECLIM, Université catholique de Louvain, B-1348, Belgium

*Correspondence to*: K. Van Oost (kristof.vanoost@uclouvain.be)

**Abstract**

Over the last few decades, soil erosion and carbon redistribution modelling has received a lot of attention due to large uncertainties and conflicting results. For a physically based representation of event dynamics, coupled soil and carbon erosion models have been developed. However, there is a lack of research utilizing models which physically represent preferential erosion and transport of different carbon fractions (i.e. mineral bound carbon, carbon encapsulated by aggregates and particulate organic carbon). Furthermore, most of the models that have a high temporal resolution are applied to relatively short time series ($< 10$ yr$^{-1}$), which might not cover the episodic nature of soil erosion. We appl~~iedy~~ ~~an~~ the event-based multi-class sediment transport (MCST) model to a hundred year time series of rainfall observation. The study area wa~~i~~s a small agricultural catchment (3 ha) located in the Belgium loess belt about 15 km southwest of Leuven, with a rolling topography of slopes up to 14%. Our ~~findings~~ modelling analysis indicates (i) that interrill erosion is a selective process which entrains primary particles, while (ii) rill erosion is non-selective and entrains aggregates, (iii) that particulate organic matter is predominantly encapsulated in aggregates, and (iv) that the export enrichment in carbon is highest during events dominated by interrill erosion and decreases with event size.

## 1 Introduction

Numerical models of soil detachment, transport and deposition are important tools for improving our understanding of soil systems and the linkages between the terrestrial and aquatic ecosystems. At present, a wide range of ~~simulation~~ erosion models are available. Conceptual models, such as the RUSLE (Römkens et al., 1997), focus largely on the prediction of long-term sediment production under various environmental and management conditions. In parallel, physically-based models have been developed to simulate the routing of soil over complex topographies, taking hydrological and sediment-sorting processes into consideration (e.g., WEPP: Nearing et al., 1989; EROSION-3D: Schmidt et al., 1991; LISEM: De Roo et al., 1996). These models operate over relatively short time scales, typically one to several events, and are concerned with modelling the detachment and movement of mineral particles. Over the last few decades, they have been instrumental in improving our understanding of erosion processes and currently serve as tools for landscape management.

Erosion-induced changes in biogeochemical cycles, in particular carbon fluxes between soils, the aquatic environment and the atmosphere, have received considerable attention over the past two decades (~~e.g.,~~ Stallard, 1998; Renwick et al., 2004; Quinton et al., 2010). However, large uncertainties and conflicting results remain (~~e.g.,~~ Lal, 2003; Van Oost et al., 2007; Kuhn et al., 2009), and this has spurred renewed interest in the application of soil erosion models. To date, few soil and carbon erosion models integrate detailed transport processes. There have been attempts to address this issue using single-point models with varying degrees of complexity (Harden et al., 1999; Manies et al., 2001; Liu et al., 2003; Billings et al., 2010). These models apply prescribed carbon erosion and/or deposition rates and simulate the resulting effects on the SOC profile using CENTURY (Parton et al., 1988) parameterizations. Recently, spatially explicit models that combine erosion models with models of carbon dynamics have been developed (e.g., Changing Relief and Evolving Ecosystems Project (CREEP): Rosenbloom et al., 2001; Yoo et al., 2005; SPEROS-C: Van Oost et al., 2005a; Fiener et al., 2015). Both CREEP and the model presented by Yoo et al. (2005) focus on long-term landscape development (i.e. millennial scale) and diffusive geomorphic processes that occur on undisturbed grasslands. The CREEP model also simulates textural differentiation and preferential transport of the finer fractions by surface wash. Compared to CREEP, SPEROS-C focuses on shorter timescales (i.e. years to decennia) and agricultural landscapes. It includes spatially distributed water and tillage erosion and dynamically couples carbon turnover, ~~puts a greater emphasis on erosion process description and emphasizes different processes, such as interrill erosion, concentrated flow erosion (e.g. rills and gully erosion), and diffusive processes due to tillage erosion~~ (Van Oost et al., 2005a; Dlugoß et al., 2012).

Although these model concepts have facilitated an improved qualitative understanding of carbon erosion and erosion-induced changes in soil carbon storage, they are largely based on unverified assumptions and simplified process descriptions: First, carbon erosion is mostly approximated as being proportional to the bulk carbon:sediment ratio of topsoils. However, both experimental and modelling studies have clearly shown that erosion preferentially removes and exports soil organic carbon (SOC; ~~e.g.,~~ Polyakov and Lal, 2004; Schiettecatte et al., 2008a b; Kuhn et al., 2010). This preferential transport results from the fact that SOC is not distributed uniformly throughout the soil, but instead consists of several fractions, characterized by

different densities and particle sizes. For example, some soil organic carbon is bound to the fine mineral fraction, some is encapsulated in soil aggregates, while another SOC fraction exists as mineral-free particulate organic carbon (POC) and has a much lower density (~~e.g.,~~ John et al., 2005; Von Lützow et al., 2007). This differentiation is particularly relevant for the C cycle, since for example the C fraction with the highest potential mobilization and transport capacity (i.e. POC due to its low density) is also a very labile fraction (Haynes, 2005). Thus, carbon erosion models should always consider the differential behaviour of sediment particles and SOC fractions when simulating erosion and transport processes. Second, carbon erosion simulation models need to consider relatively long time scales, i.e. several years to decades, as carbon erosion fluxes are relatively small when compared to rates of soil C turnover (Fiener et al., 2015). Current models addressing erosion (e.g., CENTURY: Parton et al., 1988; EPIC: Williams, 1995; WaTEM: Van Oost et al., 2000; EDCM: Liu et al., 2003) use a constant average annual soil erosion rate by assuming uniformity. However, empirical observations indicate that soil erosion and sediment delivery are to a large extent controlled by extreme events (~~e.g.~~ Fiener and Auerswald, 2007). This calls into question whether the effects of erosion on biogeochemical cycles can reasonably be derived from continuous average long-term erosion rates. Event size also influences the extent to which selective transport takes place in erosion processes. For example, interrill erosion, which is a selective process (~~e.g.,~~ Kuhn et al., 2010), is more pronounced during smaller erosion events. As a result, there is more enrichment of fine soil fractions, including carbon associated with clay particles, during small events compared to large ones. It is therefore important that carbon erosion models correctly represent the different processes that control selectivity. Furthermore, analysis on the relative contribution of low intensity (but high frequency) and more extreme (but low frequency) erosion events is required to understand the long-term effect on soil carbon dynamics. An important limitation of current approaches is therefore the frequent use of the USLE (Wischmeier and Smith, 1978) as a basis for erosion prediction. The USLE was not designed to estimate frequency distributions of soil erosion but is in fact designed to average out variability ~~(Wischmeier and Smith, 1978)~~ but is widely used on an annual (Ligonja and Shrestha, 2015, Erol et al., 2015) or monthly (Galdino et al., 2016) resolution.~~.~~

~~The Multi-Class Sediment Transport (MCST) model (Van Oost et al., 2004; Fiener et al., 2008) enables the simulation of carbon erosion, deposition and export. It also has the potential to be modified in order to address some of the issues discussed above. For example, assessing the effect of event magnitude on soil carbon erosion can be achieved by utilizing a range of rainfall input data. In addition, the MCST model uses selective transport to route sediment over areas of erosion and net deposition. Therefore, preferential export of finer particles, as well as the deposition of coarser particles, is represented by the model. Overall, by combining a model for the dynamic simulation of rainfall-runoff processes with a multi-class sediment transport model that considers different carbon fractions, the MCST allows for the examination of sediment and carbon mobilization and export in a temporal framework.~~

The main objective of this paper is to use a physically-oriented erosion model the Multi-Class Sediment Transport (MCST) model (Van Oost et al., 2004; Fiener et al., 2008) in a numerical experiment to improve our mechanistic understanding of sediment ~~redistribution~~ and carbon delivery ~~using numerical experiments~~. To this end, the MCST model is modified to incorporate the natural long-term variability of soil and soil organic carbon erosion. Existing empirical observations, covering

a century of carbon delivery, will be used to assess the model behaviour and to identify potential deficiencies in model process descriptions. Finally, the long-term role of event size on ~~both rates and patterns of~~ soil and carbon erosion will be evaluated and discussed.

## 2 Methodology

The MCST model (Van Oost et al., 2004; Fiener et al., 2008) combines a soil infiltration component with a kinematic wave routine to produce continuous series of runoff events. The event-based soil erosion component describes detachment as a function of rainfall characteristics, slope and discharge, while transport and deposition are simulated using the Hairsine and Rose (1992a, -b) equations. The two-dimensional implementation in a regular grid ($1 \times 1$ m$^2$ to $5 \times 5$ m$^2$) uses a digital elevation model to route overland flow and sediment redistribution. A detailed model description can be found in Van Oost et al. (2004)

and Fiener et al. (2008); here we focus on its main features and modifications made in order to continuously simulate long-term (up to centuries) soil and carbon erosion.

### 2.1 Modelling surface runoff

The model calculates rainfall excess at a fine temporal resolution (minutes to hours) using a modified Curve Number approach. The original version of the MCST model simulates single rainfall events but is converted into a continuous simulation model

as follows: The input of the model is a continuous rainfall series with a time resolution of 10 minutes. A rainfall-runoff event is identified as a period (i) in which rainfall depth exceeds 2 mm in 24 h (<1% of total runoff excluded) and (ii) which is separated by at least 72 h without rainfall. Accordingly, a rainfall-runoff event is not necessarily defined by a single hydrograph, but might contain multiple runoff peaks. A moving window of 24 h is used to estimate cumulative rainfall ($P_{i,cum}$) and cumulative abstractions for each time-step $i$ (i.e. initial abstraction ($I_{a,cum}$) and continuing abstraction ($F_{a,cum}$) of the curve

number method). The excess rainfall hyetograph ($R_i$) at time step $i$ is calculated as:

$$R_i = R_{i,cum} - R_{i-1,cum} \qquad\qquad and \qquad\qquad (1)$$

$$R_{i,cum} = \left(P_{i,cum} - I_{a,cum} - F_{a,cum}\right)I_f \qquad\qquad (2)$$

where $P_{i,cum}$ (mm) is the cumulative excess rainfall during the last 24 hr.

$I_f$ is a scaling factor for rainfall intensity which is calculated as:

$$I_f = \left(\frac{IN_{max10}}{10}\right)^{0.9} \qquad\qquad (3)$$

where $IN_{max10}$ is the maximum 10-minute rainfall intensity (mm/h).

Flow discharges for each grid cell and time-step are calculated by numerically solving the kinematic wave equations (Van

Oost et al., 2004). For sheet flow, cross-sectional flow area is calculated assuming a homogeneous flow depth for~~in~~ each raster

cell, while for concentrated flow, a relationship between discharge and cross-sectional flow area is used (Govers, 1992). To distinguish between sheet and concentrated flow, a critical shear velocity of 3.5 cm s$^{-1}$ for rill initiation, based on flume experiments conducted by Govers (1985), is used. The model keeps track of changes in the pattern of concentrated flow and rill network development. Finally, sediment movement is described by utilizing an event-based steady-state sediment continuity equation proposed by Yu et al. (1997).

## 2.2 Modelling erosion and deposition

Experimental research has shown that the Hairsine-Rose model provides a~~n accurate~~ physically based description of sediment transport and deposition for multiple sediment classes that differ in terms of settling velocities (Beuselinck et al., 2002a b). Transport of soil by overland flow is characterized by simultaneous re-entrainment and deposition (i.e. temporary settlement) of sediments:

$$d_i - r_{ri} = \alpha_i C_i v_{si} - \frac{\alpha_i HF}{g} \frac{\delta_i}{(\delta_i - \rho)} \frac{(\Omega - \Omega_{cr})}{D} \frac{M_{di}}{M_{dt}} \tag{4}$$

where $d_i$ is the mass rate of deposition per unit area of size class $i$ (kg s$^{-1}$ m$^{-2}$), $r_{ri}$ is the rate of sediment re-entrainment for settling velocity class $i$ (kg s$^{-1}$ m$^{-2}$), $C_i$ is the mean sediment concentration (settling velocity class $i$ ) (kg m$^{-3}$), $\alpha_i$ is the ratio of the sediment class concentration of flow ~~adjacent~~ related to the local ~~bed to the mean~~ sediment class concentration of the parent material, $v_{si}$ is the settling velocity of sediment size class $i$ (m s$^{-1}$), $H$ is the fractional shielding of the soil by the deposited layer, $F$ is the fraction of stream power used for re-entrainment, $g$ is gravity (m s$^{-2}$), $\delta_i$ is the sediment density of settling velocity class $i$ (kg m$^{-3}$), $\rho$ is the water density (kg m$^{-3}$), $\Omega$ is the stream power (W m$^{-2}$), $\Omega_{cr}$ is the critical stream power (W m$^{-2}$), $D$ is the depth of the water flow (m), $M_{di}$ is the mass of sediment class $i$ in the deposited layer (kg m$^{-2}$), $M_{dt}$ is the total mass of the deposited layer per unit area (kg m$^{-2}$).

If the local stream power ($\Omega$) is less than a critical threshold ($\Omega_{cr}$), re-entrainment does not occur and deposition of size class $i$ is a function of its specific settling velocity (Beuselinck et al., 1999; Hairsine et al., 2002). If the local stream power exceeds this threshold value, a shielding factor $H$ is calculated to decide whether net erosion or deposition occurs (Hairsine and Rose, 1992a):

$$H = \frac{(\delta - \rho)g \sum v_i C_i}{\delta F(\Omega - \Omega_{cr})} \tag{5}$$

If H $\geq$ 1, then net deposition, characterized by steady state flow and re-entrainment of previously deposited sediment, occurs. If $H < 1$, net erosion occurs, and soil detachment is modelled as:

$$D_r = aS^{ser}Q^{de} + bS^{sei} \tag{6}$$

$$D_{ir} = bI^2 Sf \tag{7}$$

where $D_r$ and $D_{ir}$ are the rill detachment rate and the interrill sediment transport to the rill (kg m$^{-2}$ s$^{-1}$), respectively, $a$ is the rill erodibility factor, $b$ is the interrill erodibility factor, $Q$ is the rill discharge (m$^3$ s$^{-1}$), $S$ is the local slope gradient, $I$ is the maximum 10 min rainfall intensity, $Sf$ is a slope factor and $ser$, $de$ and $sei$ are calibration exponents.

Rill erosion is considered to be unselective, i.e. the sediment particle size distribution of the eroded material equals the distribution of the source material at the source location. In contrast, interrill erosion is simulated as a selective process: assuming steady state flow conditions, Eq. (4) is used to estimate the particle size distribution of the sediment detached by interrill erosion and Eq. (7) is used to estimate the transport for sediment delivered to the rill network (or that leaves a grid cell when there is no incised rill). This approach is consistent with empirical observations showing that the enrichment of finer

sediment particles and SOC in suspended sediment is mainly controlled by the transport capacity of the flow (Schiettecatte et al., 2008a). To represent the amount of primary particles vs. soil aggregates of suspended sediments, the model interpolates the settling velocity for each particle class and grid cell according to the proportion of particles detached by interrill or rill erosion.

The MCST model keeps track of spatio-temporal changes in particle size distribution of the eroded and deposited topsoil

sediment within 10 different size fractions. However, the particle size distribution is spatially homogenous and constant throughout the 100 yrs. modelling period. The model considers 10 different size fractions.

**2.3 Model implementation**

For our modelling based analysis, we combined data from different sources into a virtual catchment data-set: (i) All basic data (i.e. digital elevation model, soils) were taken from a small first-order catchment in central Belgium, located about 15 km

southwest of Leuven. The site has a size of 3 ha with a mean and maximum slope of 7% and 14%, respectively. The catchment consists of diverging convex hillslopes and a central concavity where ephemeral gullying and sediment deposition are frequently observed (Fig. 1; Desmet and Govers, 1997). Soils in the catchment are loess-derived, silty-loamy Luvisols, with a clay, silt and sand content of 14%, 82% and 4%, respectively (Desmet and Govers, 1997). (ii) For the 100 yrs. modelling period, high resolution rainfall data (1899898-1997; 10-min intervals), measured in Ukkel (Brussels-Capital Region), were

used (Fig. 1, Verstraeten et al., 2006).

Modelling parameters were derived from earlier studies: (i) We assumed continuous maize cropping, where monthly Curve Number values range between 83 and 89 (Van Oost et al., 2004) to account for seasonal changes in crop cover and soil crusting. This range resulted in runoff volumes that are consistent with field observations (Gillijns et al., 2005). (ii) Two annual tillage operations are assumed to erase the network of rills and ephemeral gullies which may have evolved during preceding erosion

events. Apart from removing rills, tillage erosion is not taken into account. (iii) The rill and interrill erodibility parameter values, as well as the slope and discharge exponents (Eq. 6 and 7), were assumed to be constant over time and space. Therefore, spatio-temporal variability of soil moisture is not accounted for. The parameter values are taken from flume and plot-scale experiments, conducted using soils from the Belgium loess belt (Table 1; Van Oost et al., 2004). With these parameters, MCST

has already shown to be able to predict the spatial patterns and rates of sediment detachment and transport in the test catchment (Van Oost et al., 2004).

In simulation studies, the particle size distribution is typically derived from dispersed sediment samples and therefore reflects the settling velocities of the primary particles of the sediment. However, sediment transport and deposition can also occur in the form of aggregates, particularly for fine textured soils, as is the case in our study area (Beuselinck et al., 2000~~c~~). Therefore, we considered the particle size distributions of both aggregated soil and primary particles in our simulations. We consider~~ed~~ two erosion scenarios: ~~(i) e~~Erosion scenario 1, in which both detachment by rill and interrill erosion leads to aggregate breakdown ~~detach primary particles~~and soil is transported and deposited following the settling velocity classes of primary particles (Fig. 2). Furthermore, POM is an individual free floating particle class. ~~and (ii)~~ In erosion scenario 2, interrill erosion still breaks down aggregates and transports primary particles. In contrast, detachment by rill erosion does not lead to aggregate breakdown and entrains aggregated soil, following the settling velocity classes of aggregated soil (Fig. 2) ~~in which interrill erosion detaches primary particles and rill erosion detaches aggregates~~. For aggregated soil POM is assumed to be encapsulated in soil aggregates and is not treated as an individual class. Following detachment, the model simulates the transport and deposition of primary particles or aggregate~~d soils~~ based on the erosion type of detachment that they underwent. The particle size distributions of primary particles and aggregated soil were taken from direct measurements (n=81) in the Belgian loess belt conducted by Beuselinck et al. (1999). The grain size distribution for aggregated soil represents the relative difference between fully dispersed and non-dispersed soil in ten different diameter classes. For these classes, ~~T~~the corresponding settling velocities were calculated according to the model of Dietrich (1982), using a density of 2.6 and 1.3 kg m$^{-3}$ for primary particles and aggregates, respectively. The density of primary particles is assumed to be close to quartz, whereas a pore space of 50% is assumed for aggregates. The settling velocity distributions (Fig. 2) show that the aggregated sediments are dominated by fractions with settling velocities between $10^{-4}$ and $10^{-3}$ m s$^{-1}$, i.e. silt sized particles. In contrast, erosion scenario 1, which solely considers primary particles, shows very low settling velocities relative to aggregated sediments (Fig. 2). This results from differences in particle size between the two fractions: aggregated soils contain fewer clay and silt sized particles, because particles of this size tend to be occluded in ~~micro and macro~~ aggregates. As a result, aggregates have larger particle sizes and faster settling velocities.

The implementation of SOC characteristics in the model is based on a SOC fractionation study that was carried out with similar soils (Luvisols) from the Belgian loess belt (Doetterl et al., 2012). In the study of Doetterl et al. (2012), soil samples were taken at 11 locations along a topographic gradient, from non-eroded to eroding and depositional sites. The results showed that 85% (±10%) of the total SOC was associated with the mineral fraction (clay and silt size), while the remaining 15% (±3%) was particulate organic matter (POM~~; Fig. 3~~). ~~The POM fraction was enriched in SOC relative to the bulk soil (enrichment factor of 1.87; Doetterl et al., 2012).~~ To our knowledge, no detailed information is available on the allocation of SOC in particle size fractions from 2 to 63 µm. ~~However, others have found that 55 to 82% of mineral-bound SOC is generally associated with clay sized particles in temperate arable soils, while the remaining 18 to 45% is associated with finer silt fractions (2-20 µm; Von Lützow et al., 2007). Based on these observations~~In terms of simplicity, and given the constraints

imposed by the model structure, we considered two types of SOC for both primary particles and aggregate soil: (i) mineral-bound SOC, which represents 90% of the total and is associated with the finest sediment class ($< 2\,\mu$m) and (ii) a POM fraction, which represents 10% of total SOC and is considered a separate class in the model, with a particle size of 250 µm and a density of 1000 kg m$^{-3}$ (Doetterl et al., 2012). Hence, SOC is represented in different particle classes but the model does not account for geochemical processes.

## 2.4 Model evaluation

We evaluated the performance of the model by comparing the predicted characteristics with those that were continuously observed in the Kinderveld (250 ha) and Ganspoel (Table 2 117 ha) agricultural catchments for two observation periods of 3 years each (6 years total observation; Van Oost et al., 2005b). The two catchments are situated approx. 15 km from our study site and are larger but very similar to our site in terms of soil properties and geomorphology. We were unable to directly apply our model to these 2 agricultural catchments, as our model has high data requirements, which could not be met due to large uncertainties in input data, or in some cases because the data was simply not available. Rather than providing an evaluation on an event-basis, we evaluated the model performance by looking at the characteristics of sediment and carbon delivery, in response to a range of erosion event-sizes. This provides a first, but stringent, test of model structure and assumptions.

## 2.5 Frequency analysis

Because the time duration of a simulated runoff event may differ significantly and can contain multiple hydrographs, it is not possible to calculate event-based frequency distributions. Therefore, the event-based model results are aggregated on a monthly basis to allow a consistent analysis, and a monthly recurrence interval of soil erosion is taken as an alternative to event frequency. For an analysis of event based recurrence intervals, we follow the rainfall event definition given in section 2.1 (72 h window). Thereby, some events may contain multiple runoff peaks. The recurrence interval ($T$) is related to the frequency ($P$) with which soil erosion ($SL$) exceeds the value $X$.

$$T = \frac{1}{P(SL \geq X)} \tag{8}$$

The recurrence interval is expressed in years when $T$ is multiplied by the number of modelled years.

To calculate the frequency of exceedance, monthly soil erosion values were ranked in increasing order, and a rank $m$ is given to each modelled soil erosion event. The exceedance probability for event $m$ is given by:

$$P(SL \geq X)m = \frac{m}{n+1} \tag{9}$$

where $n$ is the total number of events during the period.

## 3 Results and Discussion

### 3.1 Rainfall/runoff

Application of the rainfall/runoff model over a period of 100 years resulted in 792 individual rainfall/runoff events. The temporal variability of rainfall events is relatively low, as more than 70% of total rainfall is associated with events with a recurrence interval of less than 1 year. Extreme rainfall events do occur, but their relative contribution to total rainfall is limited (i.e. events with a recurrence interval $\geq$ 2 years contribute less than 18% of total rainfall). The model simulates that, integrated over the period of simulation, about 10% of the total rainfall does result in surface runoff. This is consistent with field observations in the study area, where an average of 8% was reported by Steegen et al. (2000, 2001). In contrast, the simulated temporal variability in runoff is high, and events with a larger recurrence interval, i.e. $\geq$ 2 years, make up more than 36% of total runoff. The variability in runoff is higher than that of rainfall because it is controlled by multiple factors, including rainfall amount and intensity, vegetation characteristics, soil surface conditions and the presence and/or absence of a rill/ephemeral gully network at the beginning of an event.

### 3.2 Interrill and rill/ephemeral gully erosion

In the study area, erosion can be found in the mid slopes, whereas a depositional area is located in the valley bottom (Fig. 3). Interrill erosion, modelled here as a function of slope and rainfall intensity, accounts for 14% of total sediment mobilization over a 100 year period. Rainfall intensity is the main factor controlling interrill erosion and explains about 70% (p < 0.001) of the its variability. In contrast, incised (i.e. rill/ephemeral gully) erosion was modelled as a function of slope and discharge and is therefore mainly controlled by surface runoff ($R^2$ = 88%; p < 0.001). The simulated relative contribution of interrill erosion depends on the suspended sediment concentration (SSC) and the sediment delivery ratio (SDR), which is the fraction of eroded soil that is transported to the catchment outlet. For events with low values for SSC and SDR, the contribution of interrill erosion can account for up to 100% of total sediment mobilization, but this contribution declines rapidly with increasing SSC and SDR (Fig. 44). This reflects the role of event size, whereby only larger events produce significant amounts of concentrated erosion once the hydraulic threshold for rill initiation is exceeded. This large contribution of rill erosion for sediment delivery was also observed by Wang et al. 2010 in the Kinderveld and Ganspoel catchment. In a modelling study, Wilken et al. (2016) tested the effect of different rill initiation ehraeteristics characteristics on carbon delivery in a catchment of similar loess derived soils. The results showed that rill erosion widely controls sediment and carbon delivery in catchments with high connectivity.

### 3.3 Sediment and carbon mobilization and export under different scenarios

We evaluated two erosion scenarios where different assumptions about particle size distribution are made. In erosion scenario 1, where soil is transported and deposited as primary particles and POM is an individual class and erosion scenario 2 where rill erosion detaches, transports and deposits aggregated soil and POM is encapsulated in soil aggregates (see section 2.3). sediment is detached as primary particles, whereas in erosion scenario 2, soil is detached and transported in aggregated

form during rill erosion. The simulated long-term enrichment ratio of the deposits for the fine fraction (< 2 μm), which results from selective erosion and transport and deposition processes, was found to be 0.02 and 0.6 for erosion scenario 1 and 2, respectively. For erosion scenario 1, this implies that the deposition of clay particles and POM is virtually non-existent and also suggests a very efficient export of clay materials minerals and POM from first-order catchments. However, these results are not consistent with field observations. Data derived from the Belgian Soil Map (Baeyens, 1959) shows only small differences between the primary particle size distributions of colluvial and non-eroded agricultural soils in our the study area. The reported enrichment ratio for the clay fraction of colluvial soils is 0.8 (Baeyens, 1959). The colluvial sediment is thus only slightly depleted in clay when compared to the source material. Based on this analysis, we consider the results of the simulations for erosion scenario 1 to not be physically valid. In contrast, the results of erosion scenario 2 are qualitatively similar to the observations (Baeyens, 1959), which suggests that erosion scenario 2 more accurately depicts erosion processes in our study area. This implies that the assumptions made for erosion scenario 2 are appropriate, i.e. interrill erosion detaches and transports primary particles, whereas rill erosion is unselective and detaches and transports soil aggregates. The concept of particle size-selective interrill and non-selective rill erosion which detaches and entrains the entire soil matrix was documented in numerous studies (Kuhn et al., 2010, Polyakov and Lal, 2004, Quinton et al., 2001, Schiettecatte et al. 2008a). Following non-selective splash erosion (Poesen and Savat, 1980, Poesen, 1985, Parsons et al., 1991), selectivity is caused by particle size specific deposition differences, where coarser and heavier particles settle out earlier than finer and lighter particles (Schiettecatte et al. 2008b). The model tends to slightly over-estimate the export underestimate the deposition of the finest fractions (enrichment of colluvial sediments: 0.6 model versus 0.8 field observations). Beuselinck et al. (2002b) also observed this over-estimation while confronting the Hairsine-Rose deposition theory with laboratory experiments and field data. There are a few potential explanations: (i) The settling velocity distribution does not adequately represent the physical properties of soil aggregation. (ii) The Hairsine-Rose theory does not appropriately predict the depositional behaviour of the fine fractions. (iii) Model simulations overestimate selective interrill erosion.

The clay enrichment ratios at the outlet of the catchment (i.e. clay exported sediment / clay source material) for the simulated events range between 1 and 4.8 (Fig. 55). These ratios are strongly related to the SSC: high enrichment ratios occur when SSC is low, i.e. during small events with a low recurrence interval. In contrast, low enrichment ratios (i.e. close to 1) are associated with events characterized by a high SSC. These findings are in line with other studies (e.g., Schiettecatte et al., 2008a, b, Wang et al., 2010) and emphasize the importance of event size. The contribution of interrill erosion is higher for small events and, since interrill erosion is modelled as the detachment and export of individual sediment particles, this results in a higher clay enrichment ratio. Vice versa, the contribution of interrill erosion is small for large events, resulting in enrichment ratios close to 1, since concentrated erosion is assumed to be unselective. The simulated range of enrichment ratios and the relationship between those ratios and SSC are both very similar to that which was observed at the Kinderveld and Ganspoel catchment (Fig. 55; Wang et al., 2010).

However, over a simulation period of 100 years, the flux-weighted predicted clay enrichment ratio in exported sediments was found to be 1.4, which is lower than the field observed ratio of 1.5 - 2.6 for a 6 year period for the Ganspoel and Kinderveld

catchments (Wang et al., 2010). ~~We assume that this discrepancy results from difference in sedimentological connectivity, whereas a cascade of selective erosion and deposition processes in the larger catchments lead to stronger enrichment in the delivered fines.~~ In contrast, an ~~earlier studiesy utilizing~~applying ~~the~~ MCST ~~model~~in catchments of similar scale (0.7 and 3.7 ha; Fiener et al., 2008) showed a good representation ($R^2$ ~~=~~ 0.93) of the modelled transport of fines compared to 8 years of ~~measurements~~observations ~~in two catchments. Therefore, the discrepancy between the evaluation observations (Ganspoel and Kinderveld catchments) and modelled export of fines in the study area, remains to some extent uncertain.~~

~~Possible explanations for this discrepancy are (i) that~~ aggregate breakdown during transport is an important process which is not represented by the model ~~the model underestimates the detachment and/or breakdown of aggregates into finer fractions during transport, or (ii) that concentrated erosion also results in selective detachment of sediments. (iii) However, the most likely explanation are connectivity differences between the field scale of our study area and the landscape scale catchments studied by Wang et al., 2010. Nonetheless, it is not unreasonable that the much longer simulation period resulted in lower clay enrichment ratios, since the contribution of extreme events, with high export rates but low enrichment ratios, is more important over longer time scales.~~

The simulated enrichment of carbon is directly related to the preferential export of the clay fraction through ~~by~~ interrill erosion. The simulated carbon enrichment ratios are higher than the clay enrichment ratios previously discussed and range between 1 and 9 (Fig. 4~~5~~). Exported sediment is more enriched in carbon than it is in clay due to the fact that the clay fraction is itself enriched in carbon relative to the bulk soil. The simulated relationship between SSC and carbon enrichment is similar to what was found for clay enrichment, i.e. enrichment is higher when SSC is low. This is again consistent with the Kinderveld and Ganspoel field observations (Wang et al., 2010).

It should be noted that the enrichment of exported clay and carbon was simulated assuming that interrill erosion resulted in the detachment of primary particles while concentrated erosion resulted in the detachment of aggregates. Alternatively, clay and POM fractions could be considered as individual classes in the model. However, due to very low settling velocities, nearly the entire mobilized clay and POM fractions are exported from the catchment when this is simulated. This is not in line with field observations or with experiments that show that the transport of fine-textured sediments mainly occurs in the form of aggregates (~~e.g.,~~ Beuselinck et al., 2000~~c~~, Wang et al., 2013).

## 3.4 Frequency and magnitude of erosion and delivery of soil constituents

Based on the 100 year modelling period, we analysed the effect of event based frequency and magnitude of erosion on mobilisation and delivery of bulk sediment, clay and SOC ~~on a monthly basis~~ (Fig. 6~~6~~). We found that for within catchment erosion, a large number of relatively small events ~~(~~recurrence interval $< 1.5$~~4~~. yrs.) accounts~~s~~ for about half of the cumulative erosion, while larger event~~s~~ ($> 10$ yrs. recurrence) account for only about 15%.

The SDR ~~sediment delivery ratio (SDR), which is the fraction of eroded soil that is transported to the catchment outlet,~~ was 0.18 over the 100 year simulation period, while the mean erosion rate was 12.5 Mg~~t~~ ha$^{-1}$ yr$^{-1}$. Figure 5~~6~~ clearly shows that larger events play a more important role in determining SDR than they do in determining soil erosion. Approximately 57~~9~~%

of the total sediment delivery comes from events with a recurrence interval less than or equal to 10 years (Fig. ~~6~~6). This is explained by the fact that sediment delivery is not linearly related to runoff amount: once the hydraulic threshold is exceeded (i.e. an extensive network for concentrated flow is established~~, and the rill/ephemeral gully network is directly connected with the outlet of the catchment~~) the sedimentological connectivity is highly enhanced and ~~sediment delivery rates~~SDRs can be

very large. The simulation of ~~this key mechanism~~hydrological and sedimentological connectivity requires the introduction of (i) differentiated hydrological behaviours for sheet and concentrated flow~~, and~~. (ii) rill/ephemeral gully network development tracking and (iii) the rill/ephemeral gully network connectivity to the outlet of the catchment. Our simulations show that the highest export rates occurred when the rill/ephemeral gully network was already well established at the beginning of an event. The important role of a rill/ephemeral gully network for the catchment connectivity was also pointed out in other studies

(López-Vincente et al., 2013, 2015). However, structures which interrupt the rill/ephemeral gully network potentially reduce the sedimentological connectivity to the outlet and reduce the SDR substantially (Wilken et al., 2016).

The importance of event size for simulating clay and SOC delivery is also shown in Figure ~~6~~5. Compared to bulk sediment, the delivery of clay and SOC is less driven by rare, large event s, since small events with more interrill erosion already deliver

relatively large amounts of clay and SOC. In general, the model results underline the importance of a more process based analysis of SOC redistribution, as the effects of small erosion events, e.g. upon aquatic ecosystems, are underestimated when modelling only mean bulk erosion rates.

In order to qualitatively evaluate our predicted temporal patterns for sediment delivery, we compared our results to studies that continuously measured export from small catchments. In one such study, which was carried out in small agricultural

catchments in the Belgian loess belt, Steegen et al. (2000) measured sediment delivery over a 3 year period in two first-order streams. The authors found that a single event contributed to more than 40% of the total sediment delivery during the observation period and that the sum of rare and extreme events accounts for 46%. Even more extreme results were reported from a small loess catchment (3.7 ha) in Southern Germany, where a short-term series of single runoff events accounted for up to 67% of total sediment export (> 0.5 mm runoff) over an eight-year period (Fiener et al., 2008). Although a quantitative

comparison of the model results with these empirical observations is not possible, as empirical observations in central Europe typically cover far fewer than 100 years, this analysis strongly indicates that the mechanisms incorporated into the MCST model (i) allow for a quantitative representation of the relative importance of both small and large events and (ii) account for event size related ~~spatio-temporal variability of both soil erosion and~~ sediment and carbon delivery~~distributions~~.

## 4 Conclusions

In this study, we incorporated preferential erosion and transport of sediment and soil organic carbon (SOC) fractions into a numerical model of surface runoff and sediment transport. In doing so, we were able to predict the export of these different classes of sediment and SOC from small hilly watersheds, located in a temperate region with fine-textured soils. The model

predictions were only consistent with field observations when (i) interrill erosion was simulated as a process that entrains primary particles, (ii) rill erosion is unselective and (iii) low-density POM is encapsulated within soil aggregates and cannot be entrained by interrill erosion. These results suggest that carbon enrichment at the outlet of small watersheds occurs as a result of the selective interrill transport of clay and fine-silt associated carbon. Based on the application of the model over a period of 100 years, we conclude that sediment ~~export~~ delivery is a highly episodic process. Our results show that 63% of the total sediment ~~export~~ delivery was caused by 20 single events with a rainfall recurrence > 5 years. This highlights the need to consider sufficiently long timescales when addressing the impact of lateral fluxes of sediment and nutrients on soil processes. However, the dominance of large events is less pronounced in the case of soil organic carbon delivery, where only 44% of total delivery is caused by extreme events ~~(recurrence interval > 5 years)~~. This reduced importance is associated with the selective process of interrill erosion and transport. This study highlights the need for an event-based analysis of carbon erosion and delivery in order to assess the overall effect of soil organic carbon redistribution on the terrestrial carbon balance. ~~Moreover, the episodic nature of soil organic carbon redistribution is particularly important when considering the effects of SOC input to surface water bodies.~~

**Acknowledgements**

The study is financed by the FNRS (convention 2.4590.12) and is supported by the Terrestrial Environmental Observatory TERENO-Northeast of the Helmholtz Association.

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

**Tables**

**Table 1: PMain parameter descriptions and for the model setup.**

| Symbol | Description | Unit | Range/value |
|--------|-------------|------|-------------|
| Static parameters | | | |
| | Sediment density | kg m$^{-3}$ | 2600 |
| $\delta$ | Aggregate density | kg m$^{-3}$ | 1300 |
| | Particulate organic matter density | kg m$^{-3}$ | 1000 |
| $\rho$ | Water density | kg m$^{-3}$ | 1000 |
| $\Omega_{cr}$ | Threshold of re-entrainment | W m$^{-2}$ | 0.6 |
| $g$ | gravity | m s$^{-2}$ | 9.81 |
| $v_{si}$ | settling velocity for class i | m s$^{-1}$ | 2.6 10$^{-7}$ - 5.0 10$^{-3}$ |
| Dynamic parameters | | | |
| $R_i$ | excess rainfall at hyetograph at time step i | mm | |
| $P_{i,cum}$ | cumulative excess rainfall during of past 24 h | mm | |
| $I_{a,cum}$ | initial abstraction | mm | |
| $F_{a,cum}$ | continuing abstraction | mm | |
| $d_i$ | mass rate of deposition for class i | kg s$^{-1}$ m$^{-2}$ | |
| $r_{ri}$ | rate of sediment re-entrainment for class i | kg s$^{-1}$ m$^{-2}$ | |
| $C_i$ | mean sediment concentration for class i | kg m$^{-3}$ | |
| $\Omega$ | stream power | W m$^{-2}$ | |
| $D$ | depth of water flow | m | |
| $M_{di}$ | sediment mass of deposited layer for class i | kg m$^{-2}$ | |
| $M_{dt}$ | total sediment mass of deposited layer | kg m$^{-2}$ | |
| $D_r$ | rill detachment rate | kg m$^{-2}$ s$^{-1}$ | |
| $D_{ir}$ | interrill sediment transport to the rill | kg m$^{-2}$ s$^{-1}$ | |
| $Q$ | rill discharge | m$^3$ s$^{-1}$ | |
| $I$ | maximum 10 min rainfall intensity | mm h | |
| $\alpha_i$ | sediment:parent-material ratio for class i | - | |
| $F$ | stream power fraction for re-entrainment | - | |
| $H$ | shielding by deposits | - | |
| $a$ | rill erodibility factor | - | |
| $b$ | interrill erodibility factor | - | |
| $S$ | local slope gradient | - | |
| $Sf$ | slope factor | - | |

| Parameter description | Unit | Range/value |
| --- | --- | --- |
| Water density | kg m$^{-3}$ | 1000 |
| Sediment density | kg m$^{-3}$ | 2600 |
| Aggregate density | kg m$^{-3}$ | 1300 |
| Particulate organic matter density | kg m$^{-3}$ | 1000 |
| Vertical mixing coefficient | / | 1 |
| Threshold of re-entrainment | m s$^{-1}$ | 0.6 |
| Re-entrainment parameter | Pa | 0.013 |

Table 2: Area, topographic characteristics, land use and soil of the study area and the two evaluation catchments (Std: standard deviation)..

|  | Study area | Kinderveld | Ganspoel |
|---|---|---|---|
| Area [ha] | 3 | 250 | 117 |
| Elevation [m] | 12 | 61 | 39 |
| Mean slope [°] | 4.4 | 3.8 | 3.4 |
| Arable [%] | 100 | 80.5 | 76.9 |
| Forest & pasture [%] | 0 | 16.7 | 9.0 |
| Other [%] | 0 | 2.8 | 14.1 |
| Clay [%] | 14 | 7-18 | |
| Silt [%] | 83 | 70-80 | |

**Figures**

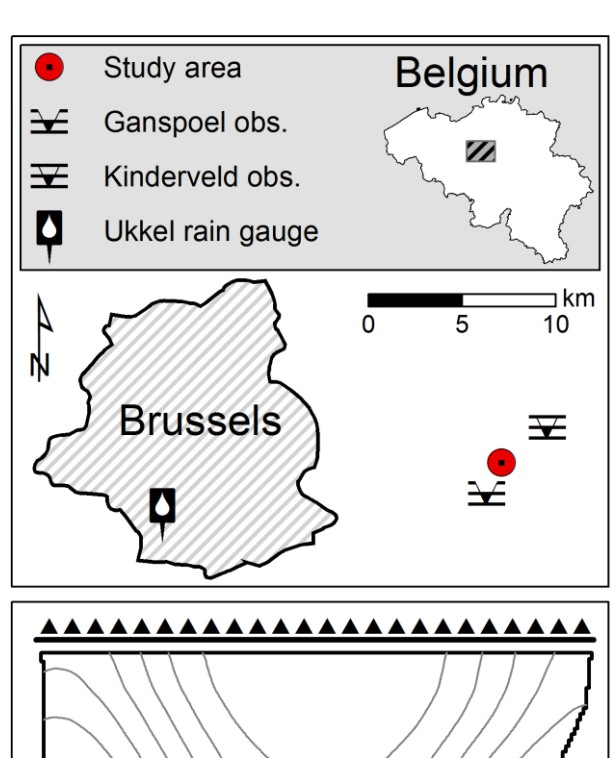

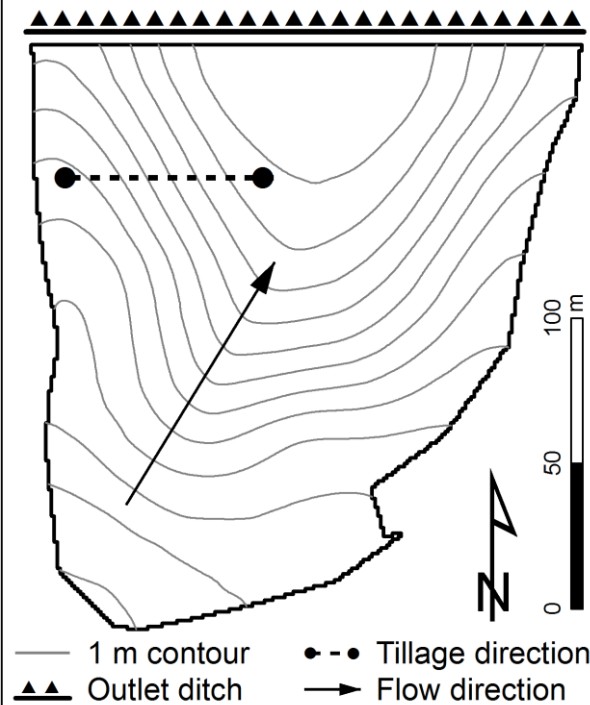

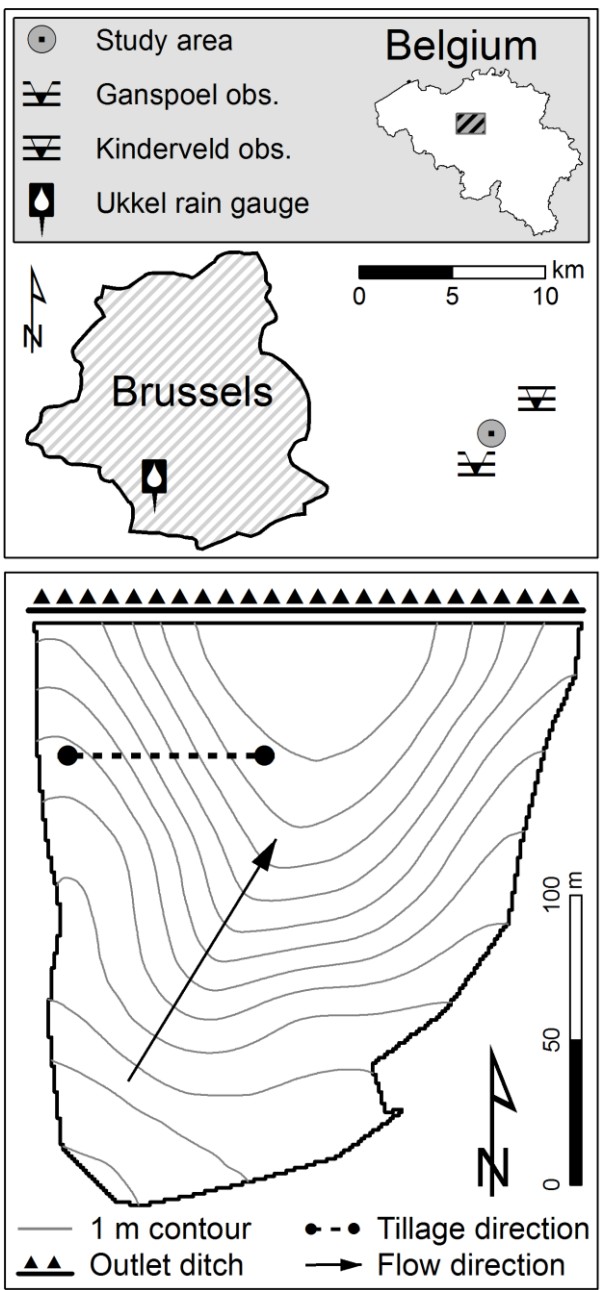

**Figure 1. Topography and location of the test catchment, location of the Ganspoel and Kinderveld runoff and sediment observation stations and the rain gauge of Ukkel, Brussels-Capital Region.**

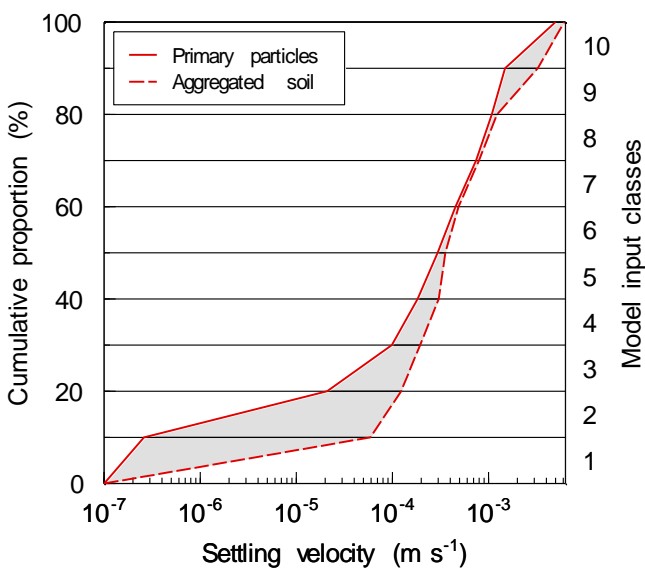

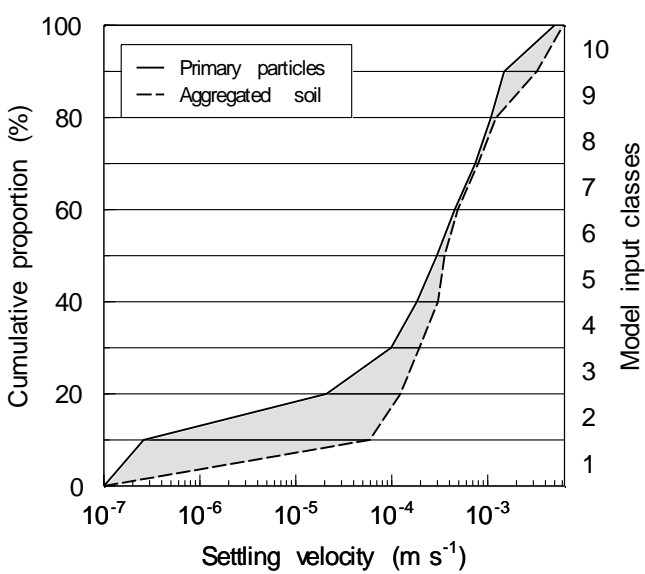

**Figure 2. Measured cumulative proportion of settling velocity distributions for primary particles and aggregated sediments (n=81). The grey area represents the range of possible settling velocities related to different proportions of primary particles or soil aggregates. Right Y-axis shows the settling velocity classes as implemented in the model for primary particles and aggregates (based on particle size distribution measurements conducted by Beuselinck et al., 1999).**

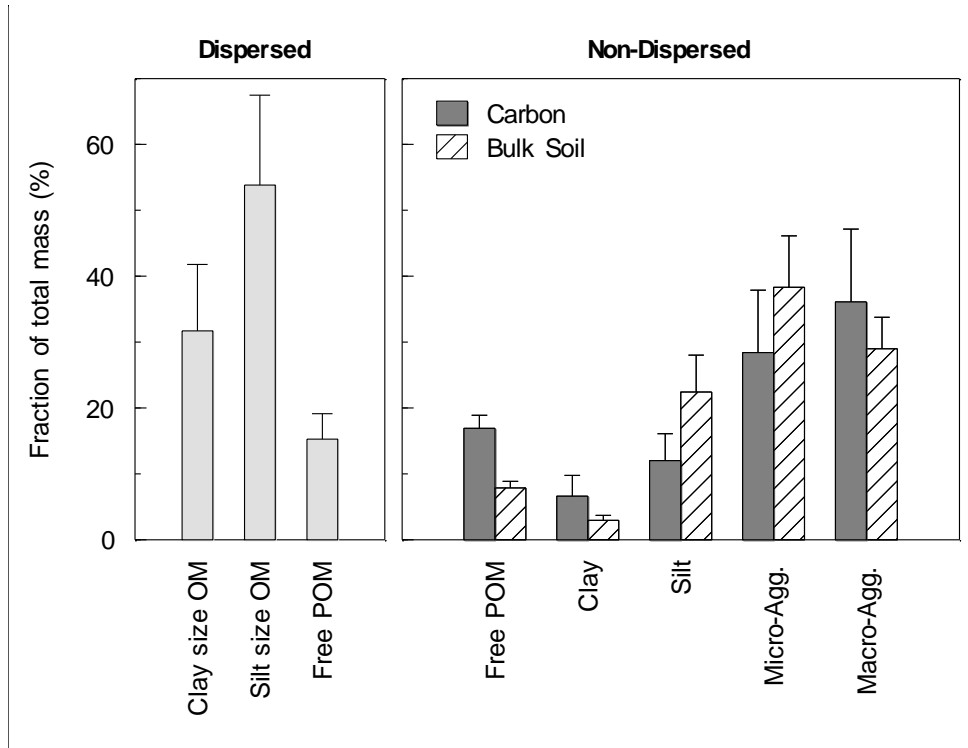

Figure 3. Results of the sediment and carbon fractionation for dispersed and non-dispersed sediments (OM: organic matter; POM: particulate organic matter).

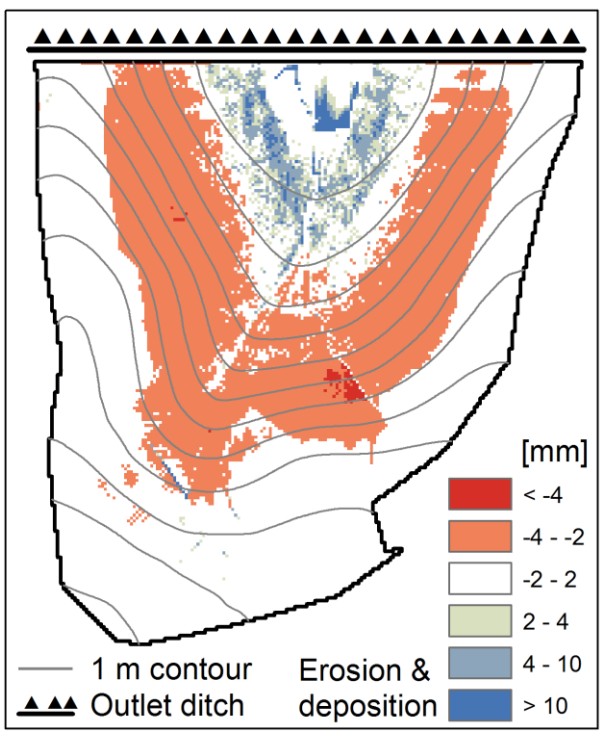

Figure 3. Spatial patterns of total soil erosion and deposition after hundred years of simulation. Negative values indicate erosion and positive values deposition.

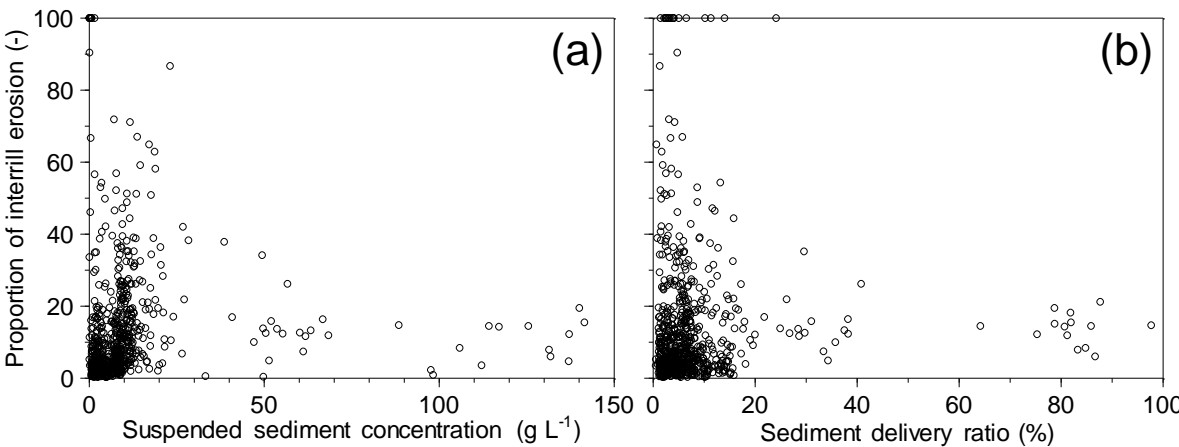

**Figure 44. Event proportion of interrill erosion contributing to suspended sediment concentration (a) and the sediment delivery ratio (b).**

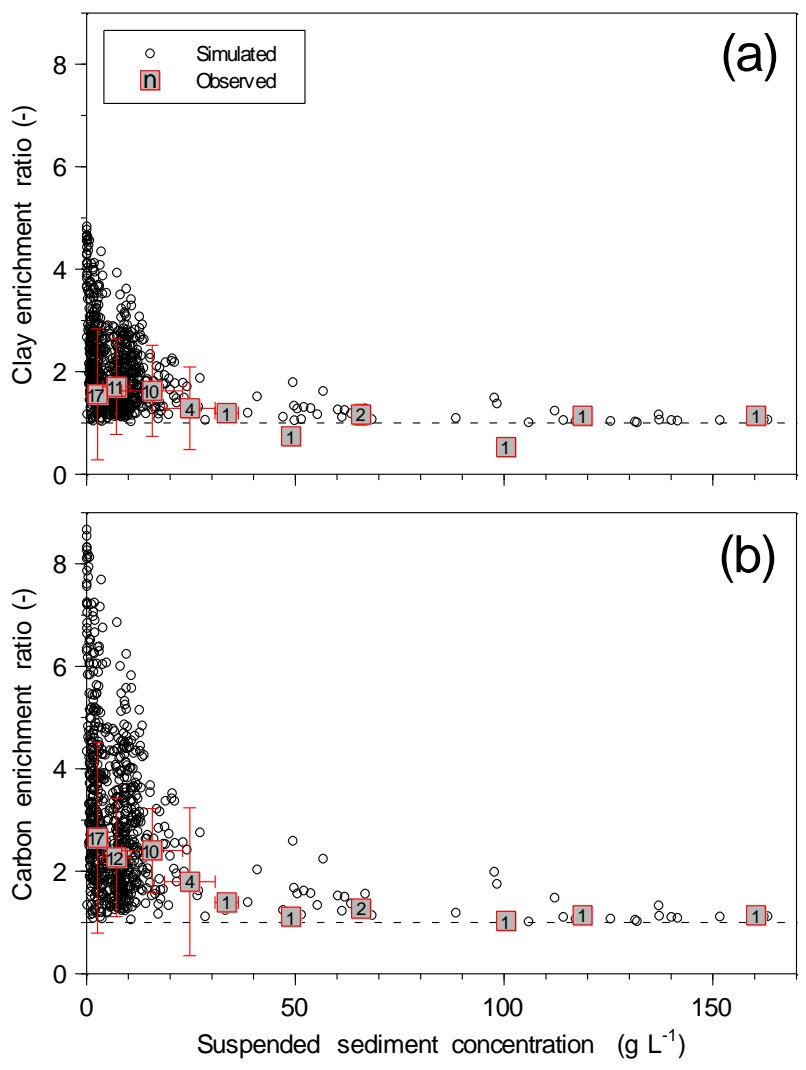

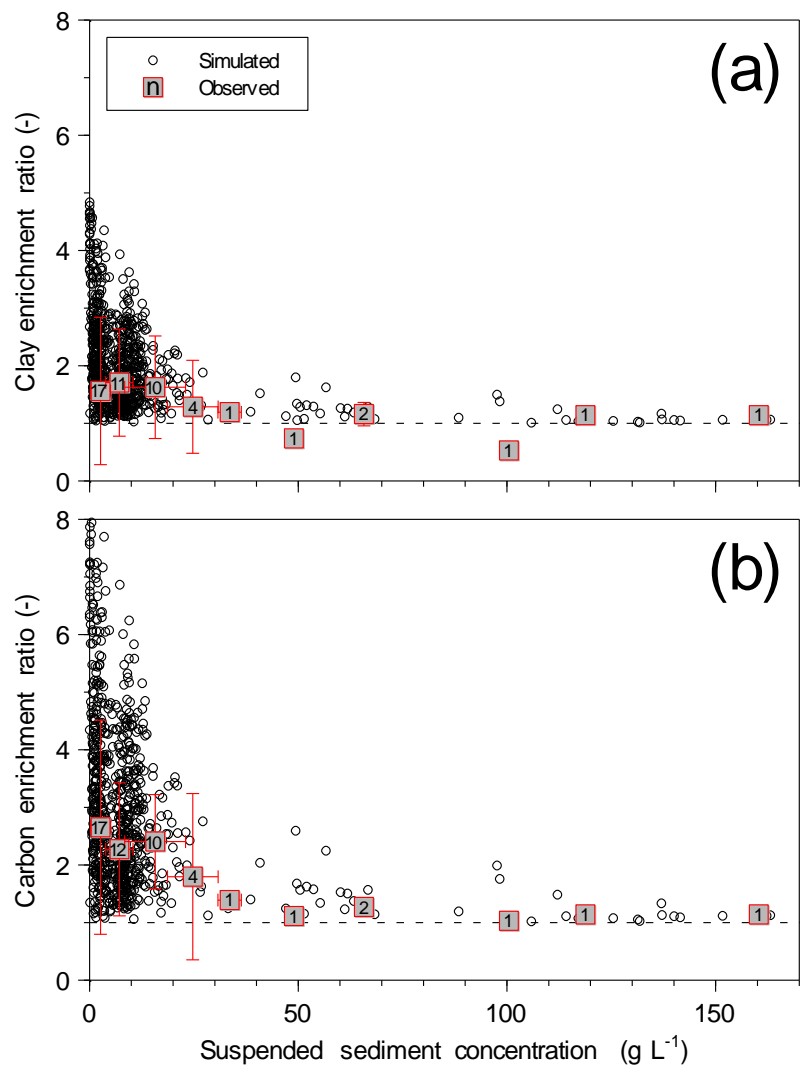

**Figure 55. Clay (a) and carbon (b) enrichment ratios with respect to simulated and observed (Wang et al., 2010) suspended sediment concentrations (observed n=clay 50/carbon 49).**

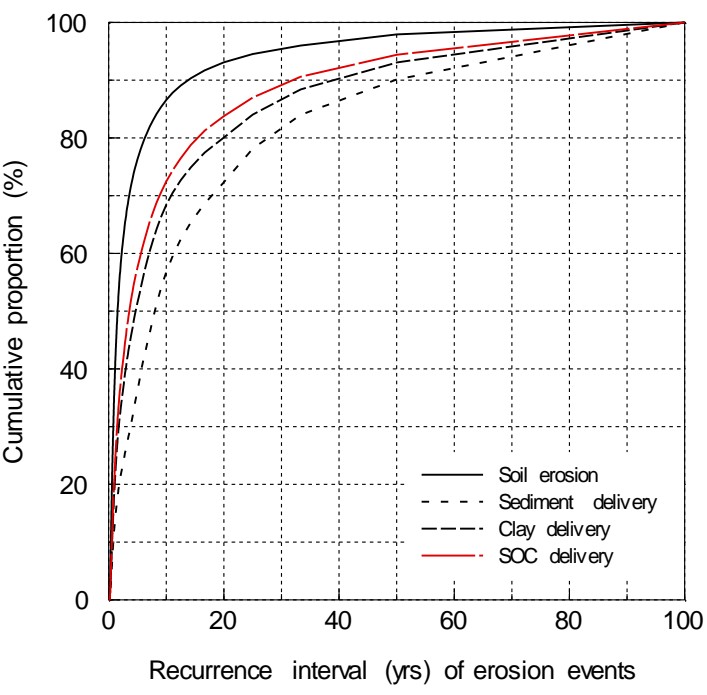

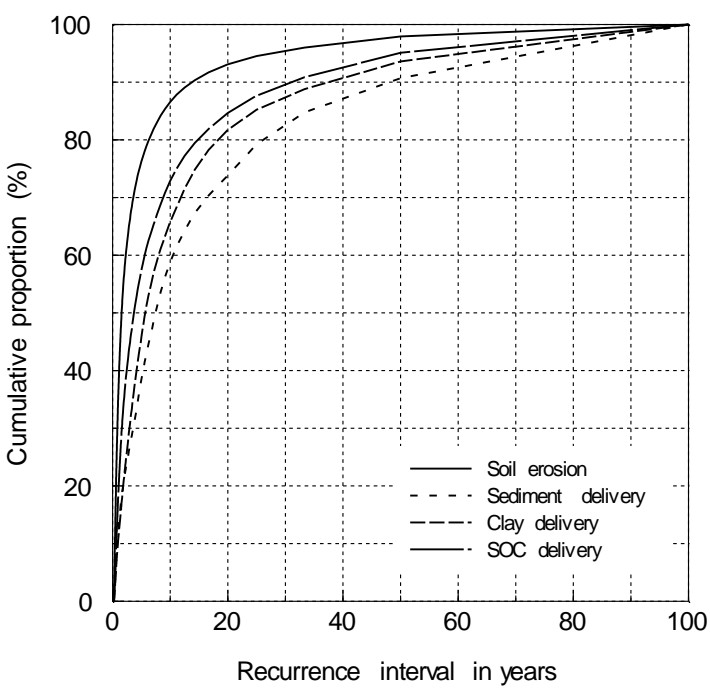

**Figure 66.** Cumulative erosion as well as sediment, clay, and SOC delivery related to event based recurrence intervals.