# Peer review of "Modelling a century of soil redistribution processes and carbon delivery from small watersheds using a multi-class sediment transport model"

_Earth Surface Dynamics, 2016_

## Referee Comment (RC1) · Anonymous Referee #1 · 26 Jul 2016

**General comments**

The paper presents a new approach where selective water erosion is studied using a model which considers episodic erosion events and selective sediment and carbon uptake. The novelty lies in this combination, and in the consideration of different types of carbon. The hypotheses and underlying theory are well explained. However, several aspects of the paper are still unclear and need extra explanation or additions: 1) the Methodology section misses explanation of some essential processes and steps. 2) the Results and Discussion section misses information on spatial patterns of sediment redistribution and requires more supporting literature. Therefore I suggest minor revisions to improve the coherence and understandability of the paper.

[Figure]

1) Methodology

- When considering longer time-scales, processes other than erosion start affecting the carbon and sediment distribution. One important process is carbon uptake. However, this is not mentioned in the paper. Please reconsider the potential effect of carbon uptake processes and include them, or elaborate on why leaving out is permitted.

- In the discussion you mention that underestimation of the breakdown of aggregates might have led to underestimation of the clay enrichment ratio (P.9, l.31-33 P.10, l.1). However, the Methods section does not mention how this breakdown is modelled. Please add a section where you explain how the model handles (micro- and macro-)aggregates, the ratio between loose sediments and aggregates, incorporation of OM in aggregates and how aggregate breakdown is organized.

- The model erases the network of rills and ephemeral gullies by tillage operations (P.6, l.19-21). However, the paper doesn't mention that tillage also induces erosion, although that can be expected in an area with steep, convex slopes. Is this incorporated, and how does this influence the results and comparison with the other catchments?

- Topographical properties like slope and catchment area have a major influence on erosion properties. Small differences in these parameters can result in different enrichment ratios and erosion patterns. Therefore I would like to see more information on the topographical and soil properties (e.g. area, slope) of the two monitored catchments which were used for model evaluation (P.7, l.22-25). You can present them in a Table, which also includes the properties of the modelled catchment.

- You aggregate run-off events on a monthly basis and use these monthly recurrence intervals for the frequency analysis (Sect. 2.5) However, I'm concerned

that with a monthly aggregation, you average out too much of the extremes and therefore too much of the erosion is attributed to smaller events. I would advise to use a smaller time-step, like 1 or 2 weeks, if the length of the longest events permit that, or add the reason why you use a monthly average. Next to that, in Sect. 3.4 you speak of events and event frequency, while you explicitly say that you will use the monthly recurrence intervals (P.8, l.1-2). Please correct this inconsistency.

- Fig. 3 is used to illustrate sediment and carbon fractionation. However, it is not clear which part of the Figure is used for this fractionation. This makes it difficult to understand what the Figure shows and how it is used in your paper. More explanation in the text or omitting an unused part of the Figure would increase its quality.

2) Results and Discussion

- The selective redistribution of carbon and sediments leads to depletion and enrichment at certain locations. These spatial patterns are in my mind one of the aspects of the objective to "improve our mechanistic understanding of sediment redistribution and carbon delivery" (P.3, l.29) and are useful for understanding variation in hydraulic properties and soil fertility. Therefore I would like to see some maps of clay and carbon redistribution in the study area and a discussion on what the consequences are of this redistribution.

- Most references in this chapter only support your results or methodology. However, I am missing references supporting your interpretation of the results. Please complement those sections with more references. Examples are the role of event size (P.8, l.26-29) and selective uptake by interrill erosion and unselective uptake by rill erosion (P.9, l. 13-15).

**Specific comments**

- You mention that "the Hairsine-Rose model provides accurate physically based description of sediment transport and deposition for multiple sediment classes that differ in terms of settling velocities" (P.5, l.4-5). However, this contradicts with P.9, l.19-20: "The Hairsine-Rose theory does not appropriately predict the depositional behavior of the fine fractions". Please consider this disadvantage in the introduction.

- P.5, l.10-11: unclear what the $\alpha$-parameter does. Can you provide a better explanation?

- P.7, l.13-14: "the POM fraction was enriched in SOC relative to the bulk soil...". Unclear what you mean by that.

- The definition of sediment delivery ratio is given on P.10, l.21. However, the term is used earlier in the paper (P.3, l.10;P.8, l.25-26). Please move the definition to one of those parts.

- P.11, l.29-30: "Moreover, the episodic nature of soil organic carbon redistribution is particularly important when considering the effects of SOC input to surface water bodies". Why is this so important and how do you conclude this from your research?

- Table 1: include a column with the symbols as they are used in the text. I also suggest to add all other model parameters and inputs (e.g. soil texture) for a complete overview.

- Fig. 2 and Fig. 3: I understood that these Figures came from other research. Please add a reference in the caption to show that the work was not carried out by you.

- Check the Methods section for past tense (e.g. P.7, l.22: "We evaluate the...", but also P.9, l.2: "assumptions ... are made" and P.1, l.18: "we apply an ...")

**Technical corrections**

- P.4, l.27: "in each raster cell". Change to: "for each raster cell"

- P.6, l.15: shouldn't this be 1898-1997 in order to reach the 100 years?

- P.19, l.6: The carbon enrichment ratios displayed in Fig. 5 are between 1 and 8. Not between 1 and 9, as is mentioned in the text.

- P.14, l.5. Reference is not in alphabetical order

---

## Referee Comment (RC2) · Anonymous Referee #2 · 19 Nov 2016

Dear authors Dear colleagues, I found your paper interesting and as a reviewer I am trying to help. As reader I wish to see your paper with slightly more discussion (now is very technical) and with a shorter conclusion section. My comments (suggestions) are attached. Congratulations for this effort to understand the erosion processes I am sure will be a very successful paper. I give in my comments some examples of papers that can help you but there are others that will be of help too. Sincerely ARtemi Cerdà

Please also note the supplement to this comment:
http://www.earth-surf-dynam-discuss.net/esurf-2016-32/esurf-2016-32-RC2-supplement.pdf

[Figure]

**ESurfD**

Interactive
comment

[Figure]

**Supplement:**

[revised manuscript text omitted]

---

## Author Comment (AC1) · 15 Jan 2017

1) Methodology

When considering longer time-scales, processes other than erosion start affecting the carbon and sediment distribution. One important process is carbon uptake. However, this is not mentioned in the paper. Please reconsider the potential effect of carbon uptake processes and include them, or elaborate on why leaving out is permitted.

*It is correct that we do not considered C turnover in this study. We focus on the longer-term dynamics of sediment, clay and soil organic carbon delivery from micro-catchments. Although we agree that there is a coupling between erosion and carbon turnover processes, it is very difficult to parameterize a model to represent this accurately. Under the absence of observations and data to constrain such a coupled model at the scales under consideration here, we think that our assumption of a spatially and temporally constant C turnover is reasonable. In order to make this clear for the reader, we have added the following sentence to section 2.3:*

*"Hence, SOC is represented in different particle classes but it should be noted that the model assumes a constant C stock that is spatially and temporally constant and therefore does not account for C turnover processes."*

In the discussion you mention that underestimation of the breakdown of aggregates might have led to underestimation of the clay enrichment ratio (P.9, l.31-33 P.10, l.1). However, the Methods section does not mention how this breakdown is modelled. Please add a section where you explain how the model handles (micro and macro-) aggregates, the ratio between loose sediments and aggregates, incorporation of OM in aggregates and how aggregate breakdown is organized.

*The model does not explicitly represent the breakdown of aggregates during detachment and transport in a physical manner. There is no observational basis to accurately parameterize a model at the spatial and temporal scales considered in this study. Instead, the model uses two end-members when considering sediment particle size distributions, i.e. an aggregated state vs primary particles. These two end-members are derived from direct measurements: i.e. from laser diffraction measurements of Beuselinck et al. 1999. The laser diffraction measurements of Beuselinck et al. (1999) represent the grain size distribution difference between fully dispersed and non-dispersed soil. The erosion process controls whether primary particles or aggregated soil is transported by interrill or rill erosion, respectively. We take from the comment of the reviewer that this was not clear. We therefore added additional information on the representation of the grain size distributions (primary particle vs aggregates). We now highlight that SOC is associated to the clay fraction of both primary particles and aggregated soil (or POM) and removed the terms micro- and macro aggregates: Section 2.2 "To represent the amount of primary particles vs. soil aggregates of suspended sediments, the model interpolates the settling velocity for each particle class and grid cell according to the proportion of particles detached by interrill or rill erosion."*

*Furthermore, we have rewritten the scenario definition to make it easier and clearer for the reader.*

*Section 2.3 "Therefore, we considered the particle size distributions of both aggregated soil and primary particles in our simulations. We considered two erosion scenarios: Erosion scenario 1, in which both detachment by rill and interrill erosion leads to complete aggregate breakdown and soil is*

*transported and deposited following the settling velocity classes of primary particles (Fig. 2).
Furthermore, POM is represented as a single particle class with a density lower than that of water. In
erosion scenario 2, interrill erosion still breaks down aggregates and transports primary particles. In
contrast, detachment by rill erosion does not lead to aggregate breakdown and entrains aggregated
soil, following the settling velocity classes of aggregated soil (Fig. 2). Moreover, POM is assumed to
be encapsulated in soil aggregates and is not treated as an individual class. Following detachment, the
model simulates the transport and deposition of primary particles or aggregated soil based on the
erosion type of detachment that they underwent."*

*Section 2.3 "In terms of simplicity, and given the constraints imposed by the model structure, we
considered two types of SOC for both primary particles and aggregate soil: (i) mineral-bound SOC,
which represents 90% of the total and is associated with the finest sediment class (< 2 μm) and (ii) a
POM fraction, which represents 10% of total SOC and is considered a separate class in the model,
with a particle size of 250 μm and a density of 1000 kg $m^{-3}$. Hence, SOC is represented in different
particle classes but the model does not account for C turnover processes."*

The model erases the network of rills and ephemeral gullies by tillage operations (P.6, l.19-21).
However, the paper doesn't mention that tillage also induces erosion, although that can be expected in
an area with steep, convex slopes. Is this incorporated, and how does this influence the results and
comparison with the other catchments?

*We fully agree with the referee that tillage erosion is an important erosion process affecting soil and
SOC redistribution. However, our paper focusses on the long-term effects of long series of water
erosion events and the effect on sediment and carbon delivery. Tillage erosion is only taken into
account as it directly effects the hydrological connectivity within the catchment by erasing a potential
rill network. To make this clearer in the text, we introduced the sentence " Apart from removing rills,
tillage erosion is not taken into account." to section 2.3.*

Topographical properties like slope and catchment area have a major influence on erosion properties.
Small differences in these parameters can result in different enrichment ratios and erosion patterns.
Therefore, I would like to see more information on the topographical and soil properties (e.g. area,
slope) of the two monitored catchments which were used for model evaluation (P.7, l.22-25). You can
present them in a Table, which also includes the properties of the modelled catchment.

*We will add the recommended information to the paper:*

|  | Study area | Kinderveld | Ganspoel |
|---|---|---|---|
| Area [ha] | 3 | 250 | 117 |
| Elevation [m] | 12 | 61 | 39 |
| Mean slope [°] | 4.4 | 3.8 | 3.4 |
| Arable [%] | 100 | 80.5 | 76.9 |
| Forest & pasture [%] | 0 | 16.7 | 9.0 |
| Other [%] | 0 | 2.8 | 14.1 |
| Clay [%] | 14 | 7-18 | |
| Silt [%] | 83 | 70-80 | |

You aggregate run-off events on a monthly basis and use these monthly recurrence intervals for the frequency analysis (Sect. 2.5). However, I'm concerned that with a monthly aggregation, you average out too much of the extremes and therefore too much of the erosion is attributed to smaller events. I would advise to use a smaller time-step, like 1 or 2 weeks, if the length of the longest events permit that, or add the reason why you use a monthly average. Next to that, in Sect. 3.4 you speak of events and event frequency, while you explicitly say that you will use the monthly recurrence intervals (P.8, l.1-2). Please correct this inconsistency.

*We see the point and will follow the argument of the referee and calculate the event based recurrence interval. Therefore, we use the definition of a rainfall event also for erosion events. Hence, some events may have multiple hydrographs, which is pointed out in the text.*

*Section 2.1 "A rainfall-runoff event is identified as a period (i) in which rainfall depth exceeds 2 mm in 24 h (<1% of total runoff excluded) and (ii) which is separated by at least 72 h without rainfall. Accordingly, a rainfall-runoff event is not necessarily defined by a single hydrograph, but might contain multiple runoff peaks."*

*Furthermore, we will shorten the description of section 2.5:*

*"For an analysis of event based recurrence intervals, we follow the rainfall event definition given in section 2.1 (72 h window). Thereby, some events may contain multiple runoff peaks."*

*The values of section 3.5 will be updated but do not show substantial differences.*

Fig. 3 is used to illustrate sediment and carbon fractionation. However, it is not clear which part of the Figure is used for this fractionation. This makes it difficult to understand what the Figure shows and how it is used in your paper. More explanation in the text or omitting an unused part of the Figure would increase its quality.

*Yes, we agree. The Figure supports the basic setup but is only partially implemented to the model. After consideration, we think the Figure might confuse the reader and will therefore be removed.*

2) Results and Discussion

The selective redistribution of carbon and sediments leads to depletion and enrichment at certain locations. These spatial patterns are in my mind one of the aspects of the objective to "improve our mechanistic understanding of sediment redistribution and carbon delivery" (P.3, l.29) and are useful for understanding variation in hydraulic properties and soil fertility. Therefore, I would like to see some maps of clay and carbon redistribution in the study area and a discussion on what the consequences are of this redistribution.

*We introduced some methodological clarification in Section 2.2. "The MCST model keeps track of spatio-temporal changes in particle size distribution of the eroded and deposited topsoil sediment within 10 different size fractions. However, the particle size distribution of the soil surface is spatially homogenous and constant throughout the 100 yrs. modelling period."*

*We did not add maps of clay and carbon contents as we found this confusing for the following reasons: Changes in proportions of clay and carbon content need to be related to topsoil (or soil in plough*

*layer). Hence, any substantial change can only be recognized in depositional areas, where the proportion of deposited material is relatively large compared to the entire plough or topsoil layer. In contrast, areas with interrill erosion, which may undergo substantial depletion of fines due to selective erosion, would show only very minor effects when enrichment/depletion is related to the topsoil or plough layer. However, to illustrate the pattern of erosion and deposition in general we added a map of total erosion and deposition in section 3.2.*

Most references in this chapter only support your results or methodology. However, I am missing references supporting your interpretation of the results. Please complement those sections with more references. Examples are the role of event size (P.8, l.26-29) and selective uptake by interrill erosion and unselective uptake by rill erosion (P.9, l. 13-15).

*The following paragraphs will be added:*

*Section 3.2 "This reflects the role of event size, whereby only larger events produce significant amounts of concentrated erosion once the hydraulic threshold for rill initiation is exceeded. This large contribution of rill erosion for sediment delivery was also observed by Wang et al. 2010 in the Kinderveld and Ganspoel catchment. In a modelling study, Wilken et al. (2016) tested the effect of different rill initiation chracteristics on carbon delivery in a catchment of similar loess derived soils. The results showed that rill erosion widely controls sediment and carbon delivery in catchments with high connectivity."*

*Section 3.3 "The concept of particle size-selective interrill and non-selective rill erosion which detaches and entrains the entire soil matrix was documented in numerous studies (Kuhn et al., 2010, Polyakov and Lal, 2004, Quinton et al., 2001, Schiettecatte et al. 2008a). Following non-selective splash erosion (Poesen and Savat, 1980, Poesen, 1985, Parsons et al., 1991), selectivity is caused by particle size specific deposition differences, where coarser and heavier particles settle out earlier than finer and lighter particles (Schiettecatte et al. 2008b)."*

You mention that "the Hairsine-Rose model provides accurate physically based description of sediment transport and deposition for multiple sediment classes that differ in terms of settling velocities" (P.5, l.4-5). However, this contradicts with P.9, l.19-20: "The Hairsine-Rose theory does not appropriately predict the depositional behaviour of the fine fractions". Please consider this disadvantage in the introduction.

*We have to thank referee #1 to point us this weak argument. After an extended discussion among the authors we came to the conclusion that our arguments, why the depletion of fines in the deposits are slightly overestimated, are all somewhat speculative. Therefore, we omitted the possible (but speculative) explanations from the text.*

P.5, l.10-11: unclear what the α-parameter does. Can you provide a better explanation?

*Yes, this was hard to understand. We hope reformulating this sentence helps to understand the α-parameter: "$\alpha_i$ is the ratio of the sediment class concentration of flow related to the local sediment class concentration of the parent material"*

P.7, l.13-14: "the POM fraction was enriched in SOC relative to the bulk soil...".

Unclear what you mean by that.

*We will shorten this paragraph and remove this sentence in the text.*

The definition of sediment delivery ratio is given on P.10, l.21. However, the term is used earlier in the paper (P.3, l.10;P.8, l.25-26). Please move the definition to one of those parts.

*Thank you, we will move the explanation to P.8, l.25-26.*

P.11, l.29-30: "Moreover, the episodic nature of soil organic carbon redistribution is particularly important when considering the effects of SOC input to surface water bodies". Why is this so important and how do you conclude this from your research?

*Yes, this opens a complete new debate. We will remove this sentence.*

Table 1: Include a column with the symbols as they are used in the text. I also suggest to add all other model parameters and inputs (e.g. soil texture) for a complete overview.

*We will extend the Table accordingly.*

| Symbol | Description | Unit | Range/value |
|---|---|---|---|
| **Static parameters** | | | |
| | Sediment density | kg m$^{-3}$ | 2600 |
| $\delta$ | Aggregate density | kg m$^{-3}$ | 1300 |
| | Particulate organic matter density | kg m$^{-3}$ | 1000 |
| $\rho$ | Water density | kg m$^{-3}$ | 1000 |
| $\Omega_{cr}$ | Threshold of re-entrainment | W m$^{-2}$ | 0.6 |
| $g$ | gravity | m s$^{-2}$ | 9.81 |
| $v_{si}$ | settling velocity for class i | m s$^{-1}$ | 2.6 10$^{-7}$ - 5.0 10$^{-3}$ |
| **Dynamic parameters** | | | |
| $R_i$ | excess rainfall at hyetograph at time step i | mm | |
| $P_{i,cum}$ | cumulative excess rainfall during of past 24 h | mm | |
| $I_{a,cum}$ | initial abstraction | mm | |
| $F_{a,cum}$ | continuing abstraction | mm | |
| $d_i$ | mass rate of deposition for class i | kg s$^{-1}$ m$^{-2}$ | |
| $r_{ri}$ | rate of sediment re-entrainment for class i | kg s$^{-1}$ m$^{-2}$ | |
| $C_i$ | mean sediment concentration for class i | kg m$^{-3}$ | |
| $\Omega$ | stream power | W m$^{-2}$ | |
| $D$ | depth of water flow | m | |
| $M_{di}$ | sediment mass of deposited layer for class i | kg m$^{-2}$ | |
| $M_{dt}$ | total sediment mass of deposited layer | kg m$^{-2}$ | |
| $D_r$ | rill detachment rate | kg m$^{-2}$ s$^{-1}$ | |
| $D_{ir}$ | interrill sediment transport to the rill | kg m$^{-2}$ s$^{-1}$ | |
| $Q$ | rill discharge | m$^3$ s$^{-1}$ | |
| $I$ | maximum 10 min rainfall intensity | mm h | |
| $\alpha_i$ | sediment:parent-material ratio for class i | - | |
| $F$ | stream power fraction for re-entrainment | - | |
| $H$ | shielding by deposits | - | |
| $a$ | rill erodibility factor | - | |
| $b$ | interrill erodibility factor | - | |
| $S$ | local slope gradient | - | |
| $Sf$ | slope factor | - | |

Fig. 2 and Fig. 3: I understood that these Figures came from other research. Please add a reference in the caption to show that the work was not carried out by you.

*Of course. Thank you for this comment.*

Check the Methods section for past tense (e.g. P.7, l.22: "We evaluate the. . .", but also P.9, l.2: "assumptions . . . are made" and P.1, l.18: "we apply an ...")

*Thank you! We will correct this.*

Technical corrections

P.4, l.27: "in each raster cell". Change to: "for each raster cell"

*Thank you! We will correct this.*

P.6, l.15: shouldn't this be 1898-1997 in order to reach the 100 years?

*This would have been a really unpleasant typing error. Thank you!*

P.19, l.6: The carbon enrichment ratios displayed in Fig. 5 are between 1 and 8.

Not between 1 and 9, as is mentioned in the text.

*The text was correct. There was a mistake in the displayed y-axis and will be corrected (see below).*

[Figure]

P.14, l.5. Reference is not in alphabetical order

*We will correct this.*

---

## Author Comment (AC2) · 15 Jan 2017

P. 3, l. 18: No necessary two citations (one is enough in my opinion)

*We will remove the first citation.*

P. 3, l. 20: I agree with the complains about the use of the USLE, but this was not the mistake of Wischmeier and Smith, it was because an abuse and misuse of the methodology. I suggest to show some examples of this see here some recent examples that can support your rationale. Suggestions: Galdino et al. 2016, Erol et al. 2015, Ligonja and Shrestha 2015

*It was not our intention to criticise Wischmeier and Smith. We will include the given examples.*

P. 3, l.21-28: I suggest to move this to the method's section

*We will follow the suggestion.*

P. 10, l. 15: In my opinion the e.g. abbreviation is overused along the paper and can be removed as does not add any relevant information.

*We will remove the e.g. at all non-relevant positions.*

P. 10, l. 22: I suggest to use Mg as abbreviation. It is more and more used and makes easier the reading.

*We will move to Mg*

P. 10, l. 29: This is a key issue to understand the soil erosion processes and this is called connectivity by the scientific community. I think your discussion section can be enriched with a couple of paragraph discussing the importance of the connectivity of flows and sediments see here some papers that can help there is more literature about this that can make your paper attractive for a large community of scientists working on this issue. Recommended background literature: *López-Vicente et al. 2015, Masselink et al. 2016, Marchamalo et al. 2016*

*Thank you, we will introduce the connectivity to the discussion.*

*Section 3.2 "This large contribution of rill erosion for sediment delivery was also observed by Wang et al. 2010 in the Kinderveld and Ganspoel catchment. In a modelling study, Wilken et al. (2016) tested the effect of different rill initiation chracteristics on carbon delivery in a catchment of similar loess derived soils. The results showed that rill erosion widely controls sediment and carbon delivery in catchments with high connectivity."*

*Section 3.3 "Possible explanations for this discrepancy are (i) that aggregate breakdown during transport is an important process which is not represented by the model, or (ii) that concentrated erosion also results in selective detachment of sediments. (iii) However, the most likely explanation are connectivity differences between the field scale study area and the landscape scale catchments studied by Wang et al., 2010."*

*Section 3.4 "This is explained by the fact that sediment delivery is not linearly related to runoff amount: once the hydraulic threshold is exceeded (i.e. an extensive network for concentrated flow is established) the sedimentological connectivity is highly enhanced and SDRs can be very large. The simulation of hydrological and sedimentological connectivity requires the introduction of (i)*

*differentiated hydrological behaviours for sheet and concentrated flow, (ii) rill/ephemeral gully network development tracking and (iii) the rill/ephemeral gully network connectivity to the outlet of the catchment. Our simulations show that the highest export rates occurred when the rill/ephemeral gully network was already well established at the beginning of an event. The important role of a rill/ephemeral gully network for the catchment connectivity was also pointed out in other studies (López-Vincente et al., 2013, 2015). However, structures which interrupt the rill/ephemeral gully network potentially reduce the sedimentological connectivity to the outlet and reduce the SDR substantially (Wilken et al., 2016)."*

P. 11, l. 3-13: This last paragraph discusses the findings within the European loess belt results but probably it will be of interest to mention also the loess plateau in China, where a different environmental (biophysical and land use and management) are different. This two recent papers can help in the discussion here or in future research of the authors. Recommended literature: Zhao et al. 2016, Tian et al. 2016.

*We agree that there might be some analogies to the Chinese Loess Plateau. However, as the referee already mentioned, there are also large differences between the European Loess Belt and the Chinese Loess Plateau (e.g. slopes, land management, climate). So we prefer not to overemphasize the analogy.*

P. 11, l. 26: Already mentioned before.
*This is a mistake. We will remove the redundancy.*

General comment on figures: I suggest to use color in the figures, readers welcome color.
*We will highlight important parts of the Figures in red color.*

Figure 6: The graphs with colored lines will be easy to read.
*Yes, we will highlight SOC delivery in red color, which should help the reader.*